# Co-Designing a Citizen Science Program for Malaria Control in Rwanda

**Domina Asingizwe [1,2,\*,†], Marilyn Milumbu Murindahabi [3,4,†], Constantianus J.M. Koenraadt [4], P. Marijn Poortvliet [2] , Arnold J.H. van Vliet [5], Chantal M. Ingabire [1], Emmanuel Hakizimana [6], Leon Mutesa [1] , Willem Takken [4] and Cees Leeuwis [7]**

1   College of Medicine and Health Sciences, University of Rwanda, Kigali 3286, Rwanda;
    cingabire7@gmail.com (C.M.I.); lmutesa@gmail.com (L.M.)
2   Strategic Communication group, Wageningen University & Research,
    6700 EW Wageningen, The Netherlands; marijn.poortvliet@wur.nl
3   College of Sciences and Technology, University of Rwanda, Kigali 3900, Rwanda;
    marilyn.murindahabi@wur.nl
4   Laboratory of Entomology, Wageningen University & Research, 6700 AA Wageningen, The Netherlands;
    sander.koenraadt@wur.nl (C.J.M.K.); willem.takken@wur.nl (W.T.)
5   Environmental Systems Analysis Group, Wageningen University & Research,
    6700 AA Wageningen, The Netherlands; arnold.vanvliet@wur.nl
6   Malaria and other Parasitic Diseases Division, Rwanda Biomedical Center, Kigali 7162, Rwanda;
    ehakizimana@gmail.com
7   Knowledge, Technology and Innovation Group, Wageningen University & Research,
    6700 EW Wageningen, the Netherlands; cees.leeuwis@wur.nl
\*   Correspondence: adomina23@gmail.com
†   The first authors.

**Abstract:** Good health and human wellbeing is one of the sustainable development goals. To achieve this goal, many efforts are required to control infectious diseases including malaria which remains a major public health concern in Rwanda. Surveillance of mosquitoes is critical to control the disease, but surveillance rarely includes the participation of citizens. A citizen science approach (CSA) has been applied for mosquito surveillance in developed countries, but it is unknown whether it is feasible in rural African contexts. In this paper, the technical and social components of such a program are described. Participatory design workshops were conducted in Ruhuha, Rwanda. Community members can decide on the technical tools for collecting and reporting mosquito species, mosquito nuisance, and confirmed malaria cases. Community members set up a social structure to gather observations by nominating representatives to collect the reports and send them to the researchers. These results demonstrate that co-designing a citizen science program (CSP) with citizens allows for decision on what to use in reporting observations. The decisions that the citizens took demonstrated that they have context-specific knowledge and skills, and showed that implementing a CSP in a rural area is feasible.

**Keywords:** malaria; participatory design; co-creation; citizen science; surveillance

## 1. Introduction

Malaria remains a major public health concern in many sub-Saharan African countries, including Rwanda [1,2]. In Rwanda, a significant reduction in malaria has been achieved through the use of control measures including long lasting insecticidal nets (LLINs), indoor residual spraying (IRS), and artemisinin-based combination therapy (ACT) [2]. However, from 2012 to 2016, Rwanda experienced

an upsurge of malaria cases that was reported across the country, especially in the eastern and southern regions. This increase put the entire population at risk and children under five years old and pregnant women were the most exposed to malaria infection [3,4].

The increase of malaria in Sub-Saharan African countries urged the global community to improve the disease and vector control response because human wellbeing is one of the United Nations' sustainable development goals [5]. Since the level of investment in malaria control across the world remains inadequate [5,6], the World Health Organization supports the development of effective and locally adapted sustainable vector control [5]. The latter includes mosquito surveillance, which consists of regular reporting of the density and the pathogen prevalence rate of vectors in a specific region. This helps to identify how vectors spread the infections to hosts and to determine appropriate interventions to reduce the risk of infection [7]. In Rwanda, in addition to active surveillance of malaria cases, mosquito surveillance is carried out in 12 sentinel sites established across the country [8]. Trained entomology technicians and officers, and some trained local community members are employed on a monthly basis to undertake mosquito surveillance in their assigned areas. Hence, this requires stable financial resources for staff payment. In addition, mosquito surveillance is based on the systematic reporting of the distribution, diversity, and density of malaria vectors using pyrethrum spray and human landing catches (HLC) as mosquito collection methods. Another indicator that is being reported is the entomological inoculation rate (EIR), expressed as the number of infectious bites per person per year [9]. The entomologists submit a compiled monthly report with entomological indicators mentioned above to the person in charge of the vector control unit of Rwanda Biomedical Center for compilation, and further analysis [10] to guide the planning of interventions.

Despite this program, there are several gaps in the surveillance system. For example, beyond the 12 sentinel sites there are still many regions where mosquito surveillance is not established because of limited funds or lack of trained entomologists. Consequently, it hinders the progress in malaria reduction, and limits community awareness on malaria vectors. A possible solution to complement the current malaria mosquito surveillance is to involve the public via citizen science-based program (CSP). Citizen science as a tool for mosquito surveillance requires an understanding of who is going to collect or report what, how, and when. This paper outlines how such a surveillance program could be designed, put in place, and what preferences exist in local communities with regard to the technical and social components of such an approach. A description of what activities are required to implement such a program are also described. We focus on several aspects including (1) the process of recruiting volunteers, (2) technical tools for collecting and reporting observations, (3) frequency of collection and reporting observations, and (4) feedback generation. The following section provides the conceptual background which elaborates on existing CSPs in mosquito surveillance and explains how the co-design concept was used to develop the CSP.

## 2. Conceptual Background

### 2.1. Citizen Science as a Tool for Mosquito Surveillance

Citizen science can be described as a collaboration between scientists and volunteers, particularly to expand opportunities for scientific data collection and to provide access to scientific information for society [11,12]. With acknowledgement of participatory action research (PAR) and other community-based interventions (CBI) that have been conducted in the last decades to improve health literacy and ability to make decisions related to malaria prevention and control [13,14], citizen science has been used to actively engage people in the collection, and/or in the analysis and the interpretation of data. This approach has been explored for monitoring invasive and endemic mosquito species in developed countries [15,16]. In these countries, citizen science has provided large amounts of relevant mosquito data, hence, citizen science proved its potential in the monitoring of (invasive and endemic) mosquito species [15,16].

These projects involved volunteers that participated in different ways such as collecting and mapping *Culex pipiens* biotypes, or assessing mosquito nuisance experienced by citizens in the Netherlands (e.g., Muggenradar) [16]. Additionally, volunteers also participate in detecting country-wide changes in mosquito fauna (e.g., Muckenatlas in Germany), and adding real-time information for daily mosquito management such as the Asian tiger mosquito (*Aedes albopictus*) (e.g., Mosquito Alert in Spain) [17–19]. In most cases, these CSPs were designed to allow the volunteers to report by sharing pictures and observations online when a specific event occurred in their regions [17–19].

CSPs have become more interactive because of the availability of the Internet, and most of the projects are now online-based. However, in absence of the Internet and with limited access to electricity, for example in rural areas of Rwanda [20], traditional methods and strategies of reporting mosquito observations can be used instead (e.g., paper forms) [21,22]. In addition to providing a valuable extension of the professional surveillance networks, CSPs can have other important functions in the strategies to reduce malaria. These include the increase of public awareness and engagement in the topic (for example about mosquito-borne diseases) [23]. In addition, participation in citizen science creates new opportunities for connections between various stakeholders such as researchers, citizens, policy makers, funding agencies, and decision-makers, thereby extending their own social network [24], and it can strengthen community-based management of residual foci of malaria transmission [13].

CSPs consists of technical and social components. The technical component defines the citizen science infrastructure like the physical kit, and the technology assets. The physical kit may consist of various equipment that could be for example mosquito traps, microscopes, buildings, etc. The technology assets are the information technology-based platforms/tools and services used to collect, store, manage, process, share, visualize, and analyze information (data and metadata) which is produced by citizen science [17]. However, the technology asset is not a required component for running a CSP [25]. The social component includes the organizers of the projects or researchers, the citizens, and the social networks of connected individuals [17].

## 2.2. Co-Designing a CSP

The development of a robust and context-specific CSP for mosquito surveillance requires the inclusion of people in the design process and integration of a diverse range of experiences, interests, and knowledge [11,26,27]. Co-designing a malaria mosquito surveillance system includes defining the social structure of the program, and defining the infrastructures and the tools, sampling and feedback strategies to be used (Figure 1).

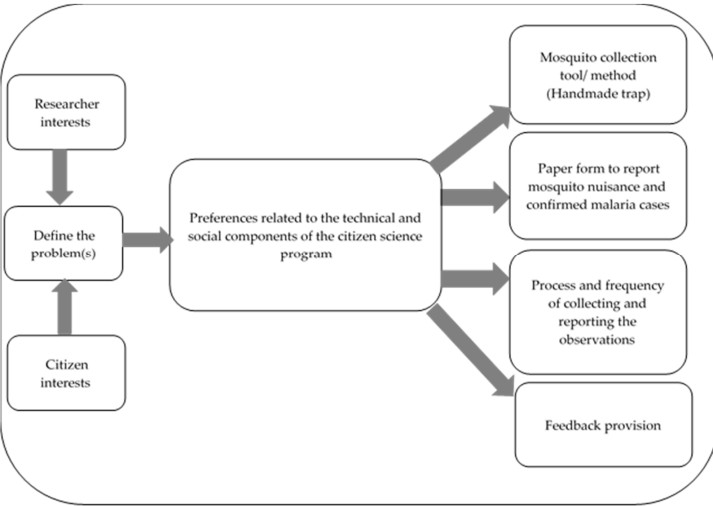

**Figure 1.** The framework indicating the co-designing process of the citizen science program for malaria control.

Participation in the design of the technical and social components of a malaria mosquito surveillance provides opportunities for the citizens to express their preferences with regard to relevant design choices, and give feedback on proposed design components. As a result, this may foster ownership of the program [17]. Additionally, citizen participation and engagement in citizen science projects increases resource capacity for mosquito surveillance on the one hand, and also promotes the acceptability of, and adherence to malaria control strategies among the community members on the other [28]. Strong partnerships with communities in vector control are required for sustainable innovation such as through citizen science [28]. The success also involves the provision of feedback that is essential in every CSP to keep people engaged in the research [27,29].

## 3. Methods

### 3.1. Study Area

The study was carried out in the Ruhuha sector of Bugesera district, located in the eastern province of Rwanda. Ruhuha covers an area of 54 square kilometers with a population of more than 24,000 (over 5500 households; 2017 data). The Ruhuha sector encompasses five cells with 35 villages. Ruhuha is bordered by Lake Cyohoha in the south and is characterized by its many water streams and marshlands classifying the region as a historical malaria endemic zone. Rice cultivation and wetland agriculture are the main economic activities. The Ruhuha sector was selected because of the high number of malaria cases reported since 2012. One village per cell was randomly selected for inclusion in the study. These five villages were Busasamana, Kagasera, Kibaza, Kiyovu, and Mubano.

### 3.2. Study Design, Population, and Sampling

Six workshops (including one for the pilot) were carried out in 2018 prior to the implementation of a CSP for malaria mosquito surveillance in selected villages in the Ruhuha sector. Participatory design workshops (PDWs) were used. A PDW is defined as a workshop through which all stakeholders including users (citizens in this case) that are affected by the upsurge of malaria in their environment, are invited to collectively define the problem that affects them, and to set up mechanisms to solve the problem while anticipating their needs [30]. It is therefore a user-centered design method in which the focus is on the active role of the users.

The first workshop was a pilot conducted in March 2018 and aimed to discuss the malaria upsurge and to explore whether participants were willing to participate in malaria control by being enrolled in the CSP and how they could participate in such a program. In addition, the pilot workshop aimed to inform the process of the main PDWs through testing the content and steps of the PDW. The pilot study was conducted in one of the ten villages in which a baseline study was conducted [28] and was randomly selected.

With the results from this pilot workshop, five follow-up workshops were organized and conducted in August 2018. Each workshop lasted around 6 hours and aimed to establish a citizen network that was willing to actively participate in the CSP.

### 3.3. Recruitment Process

Based on ten villages selected in the baseline survey [28], six of these (one for pilot and five for the main PDWs) were selected for the implementation of the CSP. Generally, each village has 150 households and we targeted a third (45 community members) of a total number of the households. In each village, the households are grouped in *isibo* (cluster of 15 neighboring households) thus each village has approximately 10 *isibos*. Therefore, three community members in each *isibo* and the *isibo* leaders were targeted to participate which results in a total of 40 participants per village. In addition, each village has three community health workers (CHWs) and one village leader. Consequently, these were also added to the 40 selected community members. Lastly, an executive of the respective cell was also expected to attend the workshop. Hence, in total 45 people were supposed to attend in each of the five

selected villages for the PDWs. This number was the same in the pilot workshop. The village leaders selected the community members and we were careful that these community members were not from the same household, or were relatives.

To ensure this, the village leader announced this workshop during a village meeting and those who showed interest were invited to participate. At the beginning of each workshop, the researchers (two first authors) verified whether the criteria have been fulfilled through requesting people from each *isibo* to stand up, and were asked whether they are from different households. Although this verification was done, however, people were not informed whether researchers were cross checking. In some few cases, it was obvious that a husband and wife could attend when one of them was a village leader and another was a CHW; this was inevitable. As shown in Table 1, in some villages community members did not attend in a sufficient numbers. The main reason for this low turn up in some villages was that these villages (Busasama and Kibaza) are located further away from the health center where the workshops were conducted. In addition, the day we conducted a workshop for Busasamana it was raining, and some community members decided to go to their farms for field work instead of attending the workshop.

**Table 1.** Characteristics of the participants attended the Participatory design workshops (PDWs).

| Village | Expected Participants | No. of Participants Attended PDWs | Male | Female |
|---|---|---|---|---|
| | | Frequency (%) | Frequency (%) | Frequency (%) |
| Busasamana | 45 | 17 (38%) | 10 (59%) | 7 (41%) |
| Kagasera | 45 | 45 (100%) | 12 (27%) | 33 (73%) |
| Kibaza | 45 | 33 (73%) | 17 (51%) | 16 (49%) |
| Kiyovu | 45 | 43 (95%) | 17 (40%) | 26 (60%) |
| Mubano | 45 | 47 (104%) | 24 (51%) | 23 (49%) |
| Total | 225 | 185 (82%) | 80 (43%) | 105 (57%) |

*3.4. Co-Design Processes*

3.4.1. Pilot Workshop

During the pilot workshop, the following guiding question was asked to the participants: "As a community member, how are you going to be engaged in malaria control?" The main raison of this question was that, "citizen engagement in malaria prevention and control activities" was one of the three strategies to improve consistent use and acceptance of malaria control measures mentioned by the participants from the baseline survey in the study area [28]. This clearly indicated that the community members were willing to participate. After the guiding question, each participant was requested to write three (maximum) ways of engagement in malaria control. These notes were then collected by the researchers, who in turn grouped those which were similar to the themes. Among the themes listed, control of mosquito breeding sites stood out. Furthermore, participation in community mobilization was also listed. Participants were divided into small groups in order to discuss the themes, and after discussion, participants presented the outcomes of the discussions. In addition, participants were requested to fill out a small questionnaire to indicate whether (1) they have ever experienced mosquito nuisance, (2) they were willing to participate in the collection of mosquitoes, and if so, (3) to describe how they think they can collect mosquitoes.

3.4.2. Five Participatory Design Workshops

The PDWs were structured in three main steps (Figure 2). As informed by the pilot workshop, the guiding questions had to be modified based on the outstanding theme. The first two authors facilitated all PDWs and the sixth co-author also joined one of the PDW.

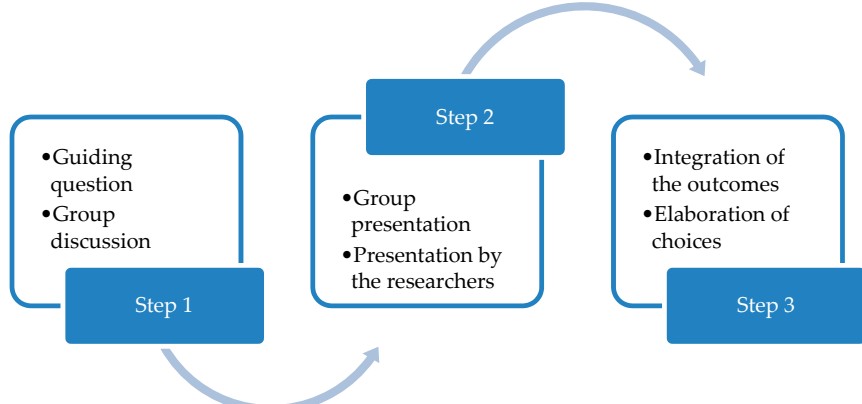

**Figure 2.** Steps followed during the participatory design workshops.

*Step 1: Part 1. Guiding Questions*

At the beginning of each PDW, guiding questions were announced. The first set of guiding questions focused on understanding of mosquito nuisance, as well as reporting of mosquito nuisance and collecting mosquito specimens: "How do you interpret the term mosquito nuisance? If you are asked to provide observations such as mosquito nuisance experienced and mosquitoes (mosquito species), how can you report it and /or collect it? Do you think it is feasible? Why would you (not) do that? How frequently can you do that"? The second guiding question focused on ways or means for providing feedback: "If you report observations, what and how would you like to get feedback"? A handout of the guiding questions was given to the groups to facilitate the discussion. It was observed that providing these guiding questions could help the participants to reflect more on their ways of engagement in malaria control rather than giving them one broad guiding question as was done during the pilot workshop.

*Step 1: Part 2. Group Discussion*

Following the guiding questions, three different homogeneous groups were formed including only men, only women, and only youth (between 18–25 years). These homogenous groups were created to ensure that all participants have equal opportunities to share their views. This was also a result of the pilot workshop, in which we observed that some participants could not express themselves in heterogeneous groups. During the focus group discussion, which lasted for 1 hour, members of the group elected a group leader and a reporter who presented the outcomes of the discussions. The different groups were requested to write down the answers of all the guiding questions on flip charts which further helped them during the presentation of the outcomes of the discussion.

*Step 2: Part 1: Group Presentation*

After group discussions on the guiding questions, a representative from each group shared their outcomes of the discussion and participants from other groups were allowed to ask questions or provide comments. After the group presentation by the different focus groups, a summary of the outcomes of each group was presented by the researchers and all participants were able to add inputs into the summary. This was done to ensure harmonization of ideas and to facilitate the integration session (Step 3).

*Step 2: Part 2. Presentation from the Researchers*

The researchers' presentation explained how to report mosquito nuisance and malaria cases by filling out a paper-based form (Figure 3a) and on how to catch and report mosquitoes (Figure 3b,c). The reporting form included two questions: (1) To what extent do you experience mosquito nuisance (unpleasant noise and biting; to be assessed on a scale from zero (no nuisance) to five (very much

nuisance))? and (2) did you have any confirmed malaria case(s) within the last two weeks in your household?

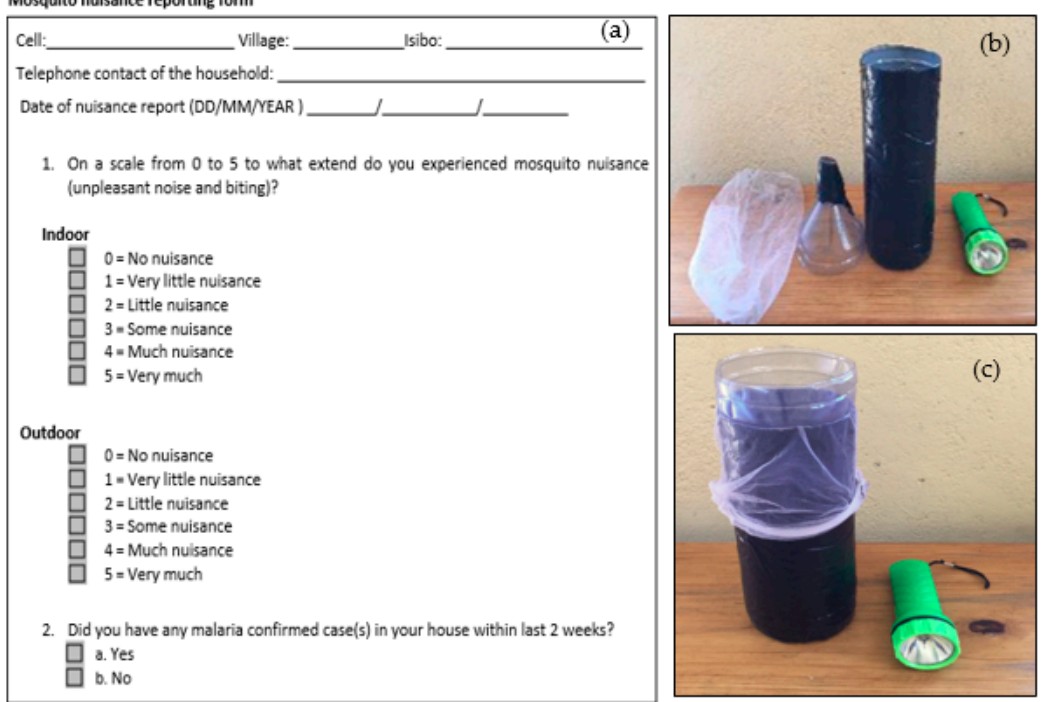

**Figure 3.** (**a**) The paper-based form for reporting observations, (**b**) the handmade carbon dioxide-baited trap showing the different elements that are part of the trap, and (**c**) the trap after assembly. The torch was suggested by volunteers and added afterward.

Additionally, participants also gave feedback on the perceived efficacy of a carbon dioxide-baited trap in collecting mosquitoes following the results presented by the researchers. This trap was prototyped and tested in the laboratory and in the field. The trap was proposed as recommended by the participants of the pilot workshop, as the majority requested tools to collect mosquitoes. Thus, researchers followed up on this by validating a low-cost, easy to use "handmade carbon dioxide-baited trap" (Figure 3c). This trap uses carbon dioxide as a stimulus which is produced by mixing yeast, sugar and water [31] and light. The trap was designed taking into account the cost and the practicability of the existing mosquito collection method in rural Rwanda. Prior to its application for the CSP, different trap designs were evaluated under laboratory conditions at Wageningen University and Research, and afterward in the field in Kibaza village (Murindahabi et al. in preparation). During the workshops, researchers presented the results from the lab and field evaluations. Furthermore, researchers presented different means of feedback provision after collecting and reporting data by the volunteers.

*Step 3: Part 1. Integration of Outcomes*

After a presentation by the researchers on the proposed tools, participants returned to their groups to discuss which proposed tools were preferred and why. Additionally, participants indicated the reporting scheme of the observations, process of reporting, preferred frequency of reporting, and ways to provide feedback. A handout of the proposed methods together with flip charts were given to the participants to facilitate the discussion and help them write down the answers for further presentation. This integration part was added to explore the different reasons to why some methods were preferred over the others and observe whether participants could be able to compare and criticize different methods including those listed by other groups. All flip charts were kept for further transcription and analysis.

*Step 3: Part 2. Elaboration of Choices*

All results from the group discussions were presented and discussed among all participants to ensure the validity of the information provided during the focus group discussions. Hence, the researchers made a summary of the choices presented by each group and then shared it with the participants for final agreement and approval.

*3.5. Data Analysis*

The results from flip charts were transcribed and analyzed. The summaries from the large group discussion taken by the researchers were also used to facilitate and inform the transcription process in case some hand writings were not readable; and compliment the transcripts in case the flip charts did not include the details because some questions for clarifications were asked during the group presentations. All transcripts were classified under different categories that emerged from the guiding questions and each group was coded independently to avoid duplication of the results. In reference to the guiding questions, different categories that included reporting mosquito nuisance, collecting mosquito specimens, recruitment of volunteers to be enrolled in citizen science, process of collecting and reporting the information, the frequency of reporting observations, and feedback generation were defined. Finally, after the completion of the data analysis, the categories were divided into two themes (technical and social) according to the co design framework of CSP (Figure 1) and the presentation of the results followed this framework as well.

*3.6. Ethical Approval*

Ethical approval was granted to the study (Approval Notice: No 414/CMHS/IRB/2017) by the Institutional Review Board of the College of Medicine and Health Sciences, University of Rwanda.

## 4. Results

The results are organized in two main sections. The first section presents the outcomes of the pilot workshop which indicate the responses about mosquito nuisance experienced and willingness to collect mosquitoes. The second section elaborates the results of the five PDWs which are divided into two main themes: (a) The technical component that includes tools to collect and report the observations, and (b) a social component that consists of (1) recruitment of volunteers, and collection of information, (2) strategies for collecting and or reporting the observations, and (3) mechanisms of feedback to the community members about the outcomes of the shared observations.

*4.1. Pilot Workshop*

Forty-four participants attended the pilot workshop. These included 29 community members, ten *isibo* leaders, three CHWs, one village leader, and one Kindama cell representative.

4.1.1. Mosquito Nuisance Experienced

All participants (100%) answered that they had experienced different levels of mosquito nuisance. Locations reported where participants experienced more mosquito nuisance are in the bush (22 times; 61%), the rice field (24 times; 69%), and near the pond (21 times; 72%).

4.1.2. Willingness to Participate in Mosquito Collections

Among the participants, 11(26%) reported willingness to collect the mosquitoes. Among these, four (36%) answered the question related to which tool/materials to be used and three (27%) were not able to mention any method. In relation to the materials to be used, one participant indicated catching mosquitoes by hand, two reported that they would need materials from researchers, and one reported using a light from a torch to "hypnotize" the mosquito and catch it afterward.

*4.2. The Participatory Design Workshops*

4.2.1. Characteristics of the Participants Who Attended the Five PDWs

One hundred and eighty-five participants (82%) out of 225 that were expected, attended the workshops (57% women; Table 1).

4.2.2. Technical Component of the CSP

In general, not many changes were made to the technical components (tools to be used to collect and report the observations) of the design as the participants already expressed that they could not report the information if the materials are not given. Below these technical tools are presented in detail.

Technical Tool to Report Mosquito Nuisance and Confirmed Malaria Cases

All groups indicated that it was feasible to estimate and report the level of mosquito nuisance experienced as well as reporting confirmed malaria cases. Participants described mosquito nuisance as the biting and the sound that the mosquito produces when flying. They all highlighted that nuisance does not necessarily relate to number of mosquitoes, because even one mosquito can bite or make noise. Among the different ways proposed by the participants to communicate the results on mosquito nuisance and confirmed malaria cases, phone calls, Short Message Service (SMS) text, and *isibo* or village meetings were mentioned. Both paper forms and SMS were viewed as a possible means to report mosquito nuisance and confirmed malaria cases. However, weighing the constraints that these two methods pose, some participants indicated that using a paper form is more preferable because it does not require much costs, while airtime credits may hinder the usage of SMS text. Therefore, the paper form was opted for to report mosquito nuisance and confirmed malaria case instead of an SMS text.

Another group of participants reported potential delays in reporting and loss of paper forms as a drawback of using paper forms. Therefore, these participants proposed to use both paper forms and SMS text or mobile phone calls as a means for reporting mosquito nuisance to the researchers. When discussing the advantages and disadvantages of using the two proposed methods, the majority of participants eventually preferred the paper form as it is less costly.

Technical Tool for Collecting Mosquitoes

Participants indicated that it was feasible to collect mosquitoes if proper mosquito sampling tools were provided. To anticipate on this and based on the results from a pilot workshop conducted in March 2018 during which the majority of the participants reported not to be able to collect mosquitoes, the researchers proposed a handmade carbon dioxide-baited traps (Figure 3c). Based on the results presented from the trap test trial (conducted in lab), participants indicated that using the carbon-dioxide-baited trap was feasible. However, some constraints with regard to the trap's practicability were mentioned. These included the difficulty to find the different components used to assemble the trap, such as the net and the 1.5 litres plastic bottle.

Other constraints mentioned by the participants were related to the costs (in case the volunteers have to buy these themselves) of the ingredients such as the yeast and the sugar used for the production of carbon dioxide to attract the mosquitoes in the trap. Participants indicated another constraint in relation to a new law that will ban the use of plastic bottles. The government of Rwanda is planning to remove plastic bottles from the market. The law on banning single-use plastics is still under evaluation in the Rwandan parliament. From the group discussions, one of the male groups indicated that catching mosquitoes using hands was also a possible option, as it does not require money.

In terms of quantity, the buckets or bottles proposed were preferred as means for collecting mosquitoes, as they seem to catch more mosquitoes than using hands and are also less sensitive to collector bias. Some of the participants reported that a bucket is easy to find, and it does not require sugar or yeast such as when using a bottle. On the other hand, if the proposed handmade

carbon dioxide-baited trap and the ingredients used as odour attractant are provided, the proposed handmade-trap using plastic bottle was the preferred tool to use by the participants to collect mosquitoes.

### 4.2.3. Social Component

Initially, the research team assumed that the data could be collected by the CHWs, or the i*sibo* leaders and can be reported on biweekly basis. Additionally, CHWs would submit the collected data to the researchers at the health center on biweekly basis as well. Alternatively, the researchers could gather the collected data at the household level in the studied villages. As indicated below, most of these initial design options changed after the discussion with the community members.

Recruitment of Volunteers for The CSP and The Collection of The Observations

There were no predefined rules on how to select volunteers and who should collect what information as all participants were eligible to participate in the program. However, as participation was on a voluntary basis, participants were invited to write their names and their telephone contacts on a list if they were willing to participate in the proposed CSP. It was made clear that they could either report mosquito nuisance and confirmed malaria cases only using the paper form, collect mosquitoes only using the proposed carbon dioxide-baited trap, or do both. It was also explained that there would be no monetary incentives for participation. Among 185 community members who attended the workshops, 116 volunteers (63%) wrote their names on the list as potential participants. Of these 116, 19 were willing to report mosquito nuisance only, 42 were willing to collect mosquitoes only, and 55 were willing to participate in both.

There was a general agreement among the participants on why it would be useful to participate in the collection and reporting of the observations such as mosquito nuisance, confirmed malaria cases, and mosquitoes. All groups highlighted that collecting mosquitoes was a way of contributing to malaria control as malaria affects many people.

Participants believed that through collecting and reporting of observations, their awareness related to mosquitoes and malaria can be enhanced. Consequently, the increased awareness may play a role in reducing malaria incidence. In addition, one of the male groups indicated that collecting mosquitoes could improve their knowledge on different mosquitoes' parts and species. However, collecting mosquitoes requires some skills and appropriate tools to do so. For this reason, one of the female groups voiced that they may not be able to collect mosquitoes because of the lack of knowledge and appropriate materials. The same concern was also reported in another mixed-gender group.

Strategies for Collecting and Reporting the Observations

The initial idea to use the CHWs, *isibo* leaders, or collecting the observation at the household level by the research team was changed because of (1) high number of *isibo* leaders (approximately 50 for the five selected villages), (2) high workload attributed for the CHWs in the community, and (3) logistic issues in case the researcher collects the data at the household level. Hence, each group of volunteers in five selected villages nominated an *isibo* representative. The organizational stucture of the reporting and collecting system was defined (Figure 4).

These *isibo* representatives were tasked to distribute the tools (paper form and mosquito collection traps), and to request and report the data on a monthly basis. They were also tasked to inform and remind the volunteers about when to fill out the forms and set up the traps. In addition, volunteers who indicated the preferred period for them to collect and report the observation in collaboration with the researchers shared a proposed schedule. The schedule indicated when to submit the collected data at the Ruhuha health center where the research team for this study was located. Hence, *isibo* representatives gathered the observations from all volunteers in their respective villages and submitted the observations to the researchers every last Friday of the month. *Isibo* representatives received feedback sent by the researchers, which in turn they shared and discussed with the volunteers among the five selected

villages. The process of reporting observations starts with distributing the tools/materials for collecting the citizen science data to *isibo* representatives, and ends when volunteers meet and discussed about the feedback from the data reported the previous month. In addition, they also discussed about the actions or measures to be taken.

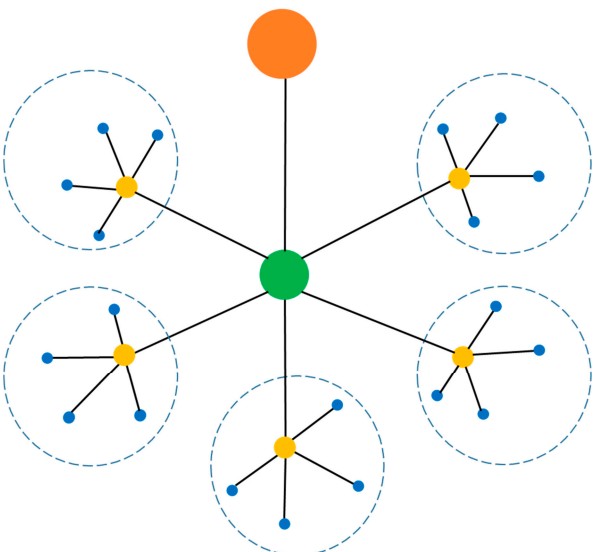

**Figure 4.** Fundamental architecture of the citizen science program (CSP) for malaria control. The blue dots represent the volunteers; the yellow dots represent the isibo representatives; the blue dashed circles represent villages, the green dot represents the researchers, the orange dot is the vector unit of the Malaria and Other Parasitic Disease Division (MOPDD). The lines represent the two-way communication between the different stakeholders involved in the CSP for malaria control. Specifically, the black lines show that these *isibo* representatives are directly connected to researchers and to the volunteers because (1) they are the ones submitting the observations every last Friday of the month, and (2) they collect the reports from volunteers, receive the feedback and share it with the volunteers in their respective villages.

Frequency of Collecting and Reporting Observations

Initially, the researchers expected that volunteers could report the observations on a bi-weekly basis. However, there was an extensive discussion on the frequency of collecting and reporting the observations because some participants assumed that collection and reporting of observations would take much time (Table 2). Hence, the general consensus was to report the observations once a month. They decided on this based on the fact that *isibo* members meet once a month.

**Table 2.** Frequency of reporting and collecting mosquitoes suggested by the different discussion groups during the workshops.

| Village | Groups | | |
|---|---|---|---|
| | **Men** | **Women** | **Youth** |
| Busasamana village | Once a month | | |
| Kagasera village | Once a month | Once in two weeks | Once a week |
| Kibaza village | Once a month | Once a month | |
| Kiyovu village | Once a month | Once a month | Once a month |
| Mubano village | Once a month | Once a month | Once in two weeks |

Note: Because of a limited number of participants for Busasamana, one mixed group was made.

Feedback Generation from the Reported Observations

Participants mentioned that getting feedback on their submitted observations was a precondition for knowing what to improve. In relation to why that feedback should be generated, participants indicated that by getting the results from what was reported, participants could be motivated to consistently use malaria preventive measures. Participants mentioned that this feedback could be given in various ways such as flyers, public talks, workshops, home visits, and SMS text messages. Participants highlighted that once submitting the mosquitoes to the researchers, they will need to know whether the collected mosquitoes were malaria vectors or not. As chosen by the participants, SMS texts could be sent to the i*sibo* and the results could be shared during the *isibo* meetings with the volunteers and during the workshop every three or four months, and via the flyers. Flyers could be distributed during the quarterly workshop as proposed by the participants.

The technical and social choices made based on the citizens' preferences are summarized in Table 3.

**Table 3.** Key technical and social choices made based on citizens' preferences.

| Components | Preferences/Choices |
|---|---|
| **Technical Design Component** | |
| Reporting mosquito nuisance | -Write down the mosquito nuisance level indoor, outdoor, and in general on paper forms and this is done every last Wednesday of the month by the volunteer.<br>-Weighing the costs of using a paper-based form and mobile phone, paper-based form does not cost much. Hence it was preferred for reporting mosquito nuisance. |
| Collecting mosquitoes | -Mosquitoes are caught with a handmade trap that consists of a plastic bottle filled with yeast and sugar, a torch, and this is done every last Wednesday of the month.<br>-Provision of materials to collect mosquitoes (yeast, sugar, torch, and the trap). |
| **Social component** | |
| Volunteer recruitment | -Everybody that attended the participatory workshop was eligible to be a volunteer. Those who were willing to participate were invited to write their names on the provided sheet. |
| Who should collect what? | -Volunteers could choose whether to report mosquito nuisance only, collect mosquitoes only, or do both. |
| Reporting the information | -Volunteers selected the *isibo* representatives who are responsible for gathering the collected information and submit them to the research team.<br>-Volunteers hand in the forms indicating the mosquito nuisance experienced and the mosquitoes caught to the *isibo* representatives during the monthly meeting that takes place in the last week of the month. The *isibo* representatives then have to submit the observations to the researchers at the health center where the researchers are based every last Friday of the month. |
| Frequency of collecting and reporting the information | -Once a month during the *isibo* meeting. |
| Feedback generation | -Once a month, researchers provide feedback to the *isibo* representative via SMS. He/she then communicates the feedback to the volunteers when volunteers collect the materials for the next round. Volunteers discuss the feedback and may take measures based on it. For example, if some malaria cases were reported, they discussed why those cases appeared and aimed to reduce the number of cases reported in the next round of reporting by a more consistent use of malaria control measures. |

## 5. Discussion

Our results indicate that a CSP for malaria control in a rural context in Rwanda is likely to work best if the inputs and insights from citizens are included in the selection of the technical and social components. By using PDWs, this study presents the design process to follow for implementation of a CSP with much attention to co-design principles as key for better implementation of a such program.

*5.1. Involving Citizens in the Co-Design Process*

Involving citizens in the design process of a CSP is of particular importance as it may stimulate other beneficial effects (for example: new knowledge) [32]. This involvement helps participants in making decision whether to participate or not. Some CSPs that do not engage citizens in the co-design process, have to incentivize participants once they submit the data [25], and these incentives (money in most of the cases) can be a key motivation for them to participate.

The co-design process that was employed in our program provided added advantages beyond normal trainings, because apart from acquiring new knowledge, the participants felt part of the design and were motivated to contribute to both scientific research as well as malaria control with no monetary incentives. The engagement of citizens in the design process may influence the recruitment rate and level of participation in CSPs [25,33]. Hence the participatory design workshops proved an important step for the community members to be able to decide whether to participate in a CSP.

When the research team started, it was optimistic about the use of mobile phones as one of the communication channels and for reporting observations. It was clear that the participants did not prefer it as this option presented more challenges than solutions to the problem. Similarly, Beza, Reidsma [34] also revealed that the price of sending SMS can affect the decision to participate in a project that require the use of mobile phone. Consequently, the mobile phones were only used in this study to provide feedback to volunteers via SMS on a monthly basis and this has no cost implications for the receiver of this SMS.

It was clear that when volunteers have different options (for example different mosquito collection tools), they can critically reflect, discuss, and decide what works better for them. Including citizens in the design process promotes critical discussion which may foster further community actions to tackle the problem under study [32]. Communities are different, have different backgrounds, ideas, and they may learn from each other. While some groups indicated that catching mosquitoes is not possible unless collection tools are provided, others proposed some materials including buckets and the use of hands to catch and submit mosquitoes to the research team. This created a new learning synergy needed to implement a CSP.

*5.2. Why Providing Feedback to Volunteers?*

Communicating the key messages that result from what people report is crucial in any CSPs [35]. Keeping volunteers updated about the progress of the project is an important aspect as it increases the interaction between volunteers and researchers, and volunteers can provide feedback on how to improve the project [35]. In turn, this feedback can retain participants, and hence sustain the project [36]. As volunteers contribute their time without financial or any other direct benefits, giving feedback is one of the non-monetary incentives that motivates participants [35]. Different forms of feedback, including automated SMS text to individual volunteers, newsletters and websites, have been used in CSPs [35,37,38]. When regular feedback is provided, it enhances opportunities for learning and development for the participants. This, in turn, may strengthen the network among participants and may improve collective practices [39]. As reported in this study, feedback provision was also considered during the design process. Participants indicated a wish to have monthly feedback from the reports in a form of SMS text. Additionally, quarterly workshops in order to meet with other volunteers and learn from others, as well as develop leaflets indicating and comparing reports from different villages were also added.

*5.3. Study Limitations and Future Research*

We realize that additional components may be needed, such as well-documented rules and regulations for guiding volunteers and preventing them using other than agreed tools. However, if tight rules are put in place, the withdrawal rate may be higher as volunteers may think that it is too difficult for volunteers if they have to obey too many rules for voluntary work. On the other hand, not putting these rules and regulations in place may lead to misuse or non-use of the agreed trap. Future research will explore the use of the trap, by assessing the quality of the reports that the citizens are submitting.

At the start, the research team was interested in the collection of mosquitoes and reporting of mosquito nuisance, and this limited the generation of information about community preferences regarding what should be observed in the first place. However, researcher motivation is in most cases an important reason to start a CSP [36]. Still, in our case there was also a clear societal reason for why to start this CSP, i.e., addressing the burden of malaria. The selection of gathering observations of nuisance and mosquitoes was based on the best available knowledge and successful experiences elsewhere [13,15]. In addition, the research team wanted to see whether participants would come up with their own mosquito collection method. Unfortunately this was not the case. Therefore, the team designed and proposed a trap that could be easily used in low resource settings. The paper-based form also was designed and proposed based on the results of the baseline survey in 2017. In this survey, only 45% of the people in the study area owned a mobile phone, and among these only a small proportion (18%) mentioned that they also use their mobile phone for SMS activities (sending and/or receiving any message) (Unpublished data). To this end, proposing a paper-based form was a way to overcome this technology-related barrier.

Besides the information gathered by the current CSP, the program may provide other information which may help to design more targeted interventions such as spraying and larval source management. Follow-up studies will determine the spatio-temporal distribution and population dynamics of the collected malaria mosquitoes in relation to malaria transmission risk, and assess how this will support the current government-led mosquito surveillance program.

The perceived initial motivations to participate in citizen science may be subject to change over time, and participants presented different motivational factors including a desire to contribute to malaria control, and to gain knowledge and awareness about mosquito species. Thus, future studies should remain exploring ongoing motivations as this is a key determinant for the retention of the participants and thus the sustainability of the program. Although in this study volunteers indicated interest to acquire new knowledge and skills, for further collective decision making it is important to evaluate the CSP by assessing throughout the participation process what people gain while participating, and whether there are some individual and collective actions (for example collective management of mosquito breeding sites [24,40] that may result.

## 6. Conclusions

Considering the possibilities and preferences of citizens prior to the implementation of a CSP for mosquito surveillance is essential for its success. Some technical as well as social changes were made together with volunteers following their preferences and choices. Following involvement of citizens in a co-design process, we arrived at different decisions that we did not always foresee beforehand. For example, involving volunteers in organizing the process of collecting and reporting observations facilitated the decision about how and who should gather the observations, and submit them to the research team. Deciding on the tools to use while reporting data is of importance, because participants know what works for them. Furthermore, this may positively influence the level of participation. The findings also revealed that providing feedback from what people reported is crucial in a CSP. Thus, this study revealed a number of technical and social components that are relevant to making a CSP applicable and feasible in rural areas and or in locations where Internet connectivity is limited. Additionally, a CSP can build the capacity and increase knowledge which in turn, may lead to further individual and collective actions for malaria prevention and control.

**Author Contributions:** All authors designed the study. D.A. and M.M.M. coordinated the study implementation, collected the data, analyzed the data, and drafted the manuscript. C.J.M.K., P.M.P., A.J.H.v.V., and C.L. provided extensive guidance in study design, data analysis, and critically reviewed the paper. C.M.I., E.H., L.M., and W.T. contributed substantially to the study implementation and revision of the paper. All authors have read and approved the final manuscript.

**Funding:** This publication is part of the project "Environmental Virtual Observatories for Connective Action (EVOCA)", project duration 2016-2020, which is funded by Wageningen University, The Netherlands through its Interdisciplinary Research and Education Fund (INREF).

**Acknowledgments:** Special thanks to the citizen science volunteers for their collaboration and involvement in this project. We also thank the Ruhuha sector leadership and Ruhuha health center for their regular support and collaboration.

**Conflicts of Interest:** The authors declare that they have no competing interests.

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
