# Peer review of "Co-Designing a Citizen Science Program for Malaria Control in Rwanda"

_sustainability, doi:10.3390/su11247012_

Round 1

Reviewer 1 Report

Thank you for the opportunity to review your paper that describes a co-designed citizen science program for malaria control in Rwanda. This is an interesting area and we are seeing increasing participation of the public in science. Understanding best practice processes to support the public to contribute to research is important and this paper provides some interesting content to help guide a citizen science approach. Overall this has the potential to be an interesting and relevant paper but it could be strengthened with some refinements. I will provide feedback under each heading of the paper as well as starting with some general feedback. Please don't be disheartened and I hope my feedback will help make this a clearer and stronger paper.

General comments

One concern I have is that there is no mention of ethical approval for this project. This is important if this work is to be published and needs to be clearly stated in the methods section (or the reason it does not require ethical approval).

The authors seem to be reporting on three separate components:
1. The design and process of the workshop and overall project (including the pilot and the five workshops, recruiting volunteers, deciding on tools etc).

The roll out (implementation) of the project. Follow up results after 12 months.

I think these three separate areas could be presented more clearly in the results section if they were treated separately.

Abstract

The abstract describes the program reasonably well. My only concern relates to the last two sentences (lines 30-31). I am not sure if the authors can say that the citizens have the context-specific knowledge and skills as no assessment of these is reported in the paper. Were these assessed after the workshops? If so this needs to be reported. It is also reported that there was a high level of participation (93%) observed throughout the first year of the project – I will comment further on this under the results heading. The authors could clearly state that the intent of this paper is to describe the design and process of delivery of the workshops, describe the implementation and the 12 month follow up results. The results could then correspond to each of these three sections.

Introduction

The introduction is well-written and provides a good context to the malaria issue in Rwanda.

Line 47 – should the references be 5-8? Otherwise there is no mention of references 6 and 7 in the introduction.

Line 75 – after point 4 it might also be worth mentioning that 12 month follow up is also reported.

Conceptual background

The process of co-design is well described and Figure 1 presents a clear framework.

Line 85 – should the references be 15-18? Otherwise there is no mention of references 16 and 17 in the conceptual background.

Line 105 – the should be their.

Methods

The study area, study design, population and sampling is clearly described. A little further detail of the pilot workshop could be provided. How was the site for the pilot selected? Where all activities presented in subsequent workshops, undertaken in the pilot, and if so what refinements were made (if any)? Did the pilot workshop actually inform the development of the content in the subsequent workshops? Lines 213-214 indicates that the trap used was recommended from the pilot workshop. It would be good to describe this and any other outcomes/changes from the pilot.

Who facilitated the follow up workshops? It is stated that community members were invited to participate (line 161-162) but on line 166 it states that community members were selected by the village leaders. Did the village leaders chose three community members out of a larger number who responded to the invitation and if so what criteria was used to select the three? What was the response to the invitation like? Did each workshops have exactly the same breakdown of participants (lines 166-170)?

Line 174 – this might just be me but I have not heard the term “calling” questions. Can the authors briefly clarify what a calling question is. It seems strange to start with a question; “as a community member, how are you going to be engaged in malaria control”, before they are asked whether they are willing to participate. Was there a reason for starting with this question?

Line 183-185 – this sentence could be reworded to make it clearer.

I am a little confused about the questions asked in the group discussion. Line 193 states “following the calling questions” and line 197 states the groups were requested to answer “all questions”. Were the questions asked to the groups the same as the calling questions described in lines 187-191 or were they different questions? Initially I thought they were different however on line 199 the authors again refer to the calling questions so maybe they were the same. It would be good to clarify that please.

Line 200 and line 201 use the term “results” – it might be better to say they shared their “discussion” rather than shared their “results”.

In the analysis section there is mention of flip charts being used – was this in the group discussion and if so how were they used/for what reason?

Line 211 – how can participants give feedback on the efficacy of carbon dioxide-baited traps in collecting mosquitoes if they had not yet started the program? It sounded like these were proposed tools (line 213) states this.

The paragraph lines 211-224 could be clearer.

Line 226 – the Figure should be labelled Figure 3 not Figure 2.

Data analysis – how the flip charts were used needs explanation. At what point did transcription occur? It sounds like they were done during the workshop and then shared with the whole group – is that correct? How was thematic analysis actually conducted? More description is required.

In the methods section I would expect some mention of ethics – was there ethical approval for this project? Or if it was not required this should be clearly stated.

Results

Overall the results presented are interesting but as mentioned it would be good to have three clear sections for the results:

The design and process of the workshop and overall project (including the pilot and the five workshops, recruiting volunteers, deciding on tools etc). The roll out (implementation) of the project. Follow up results after 12 months.

Line 257-260 (4.1.1): I was interested to see what participants listed as their expectations as these seemed to reflect they did not understand the purpose of the workshop (i.e. to get citizen science participation). I am not sure if this is that relevant to results and is probably not required. There is no further feedback as to whether their expectations were met.

4.1.2: the themes do not seem to match the heading – “How participants will be engaged in malaria control”. For example how is sleeping under bed nets and early diagnosis and treatment relevant to their engagement?

4.1.3: Why was this question relevant? Was it to determine where interventions were required?

4.1.4 is more relevant and has good content.

4.2 (line 276). I would suggest starting with the characteristics of the participants and then lead into what came out of the workshop. Can the authors clarify what is meant when they say 185 out of 225 attended and participated in the work shop. They either attend or don’t attend. Were there RSVPs perhaps from 225 which meant they were expected to attend and only 185 turned up to the workshops. This is not clear.

Line 282 – what is PDWs – I can’t see it written out in full in the manuscript but I may have missed it.

Line 311 – what is PW – I can’t see it written out in full in the manuscript but I may have missed it. Further clarity is required in the sentences on lines 311-317. The PW trial needs explanation. I am unclear if the suggestion to use the carbon dioxide baited traps came out of the pilot workshop or the PW trial.

Line 336 – no need to write community health workers out in full as this has been abbreviated previously.

From Section 4.2.3 onwards is where I think clearer delineation between the design and the roll out (implementation) of the citizen science program can be made.

On line 354 it is stated that there was 93% participation after nine months. This could come after the implementation is presented and detail about the methods to assess this are required. How was participation measured.

Line 366-367: I am surprised by the females concerns – would hey not be trained and given materials as part of being involved in the citizen science program?

Line 376 – this should be Figure 5 not Figure 4. I am not sure what Figure 4 on lines 378-394 is as it is not referred to in the text. It is also incorrectly numbered.

Lines 414-425: It is stated that reporting happened once a month but in Table 2 some report fortnightly (women in one group and youth in one group) or weekly (youth in one group).

Lines 440-441 and Table 4: I am confused where these preferences /choices came from – was it after the workshop or after the roll out of the program?

Discussion

The discussion could be clearer and may well be following some modification to how results are presented. Overall it would be good to see the main focus of the discussion on the co-design process and what can be learnt from that. Learnings from the implementation will also be of interest and this seems to covered in lines 484-497.

There is content in the discussion on line 452 and 453 that contains information that is not clearly presented in the results – i.e. hardly any variation in participation throughout the first year of the program.

I am not sure how relevant the content is that is described in lines 453-461 but rewriting may improve this clarity and relevance.

Study limitations and future research: This section could be refined.

Line 500-501: When the authors refer to well-documented rules and regulations – can these be explained. Do you mean clear guidelines/instructions?

Line 509: I am not sure if you can call motivation a key scientific reason. Just needs rewording.

I think an important recommendation should be about the need for rigorous evaluations of citizen science programs.

Conclusion

This may need to be refined following a rewrite of the results and discussion.

References

Up-to-date references used. A few minor presentation corrections required.

Author Response

Reviewer 3

General comments

#1: One concern I have is that there is no mention of ethical approval for this project. This is important if this work is to be published and needs to be clearly stated in the methods section (or the reason it does not require ethical approval).

Response:

As stated on comment #7 and #9 (under 1st reviewer’s comments), a section about ethical approval from the Institutional Review Board of the College of Medicine and Health Sciences, University of Rwanda has been added (Lines 343-345).

#2: The authors seem to be reporting on three separate components:
1. The design and process of the workshop and overall project (including the pilot and the five workshops, recruiting volunteers, deciding on tools etc). The roll out (implementation) of the project. Follow up results after 12 months. I think these three separate areas could be presented more clearly in the results section if they were treated separately.

Response:

Overall, the paper presented the results of the design process and this is the main objective of the manuscript. This design process included (1) pilot study (as it informs how the researchers decided to propose the tools for collecting the observations) and (2) five participatory design workshops as the main methods used for the design process. Apart from the process to undergo while designing CSP, we added two other results to: (1) show that program really worked, thus included the level of participation, and (2) show an example of SMS as a way of feedback provision. However, it seems that these additional results brought more confusion instead of clarity to the manuscript, so we decided to remove them. In the result section, an introduction indicating the flow of the result presentation (starting with a pilot study followed by the results of five participatory design workshops) has been added (Lines 348-354)

#3.1: Abstract : The abstract describes the program reasonably well. My only concern relates to the last two sentences (lines 30-31). I am not sure if the authors can say that the citizens have the context-specific knowledge and skills as no assessment of these is reported in the paper. Were these assessed after the workshops? If so this needs to be reported.

Response:

In these lines, the authors wanted to indicate that the citizens know what could work for them, and could participate in finding solutions. For example, some assumptions from researchers were changed as a result of the discussion with citizens. Some of these assumptions included: (1) strategies for collection and reporting the observations (lines 477-484) and (2) frequency of collecting and reporting the observation (lines 535-545).

#3.2: It is also reported that there was a high level of participation (93%) observed throughout the first year of the project – I will comment further on this under the results heading. The authors could clearly state that the intent of this paper is to describe the design and process of delivery of the workshops, describe the implementation and the 12 months follow up results. The results could then correspond to each of these three sections.

Response:

Thank you for the observation. As indicated in comment #2, the level of participation was presented to show that after the design process the actual implementation followed and we thought that mentioning this level of participation presented an added value to the design process described. However, after all, this has been deleted (see also comment #2) to improve upon the clarity of the paper.

#4: Introduction: The introduction is well-written and provides a good context to the malaria issue in Rwanda.

Response:

Thank you.

#5: Line 47 – should the references be 5-8? Otherwise there is no mention of references 6 and 7 in the introduction.

Response:

Reference numbers were corrected throughout the manuscript.

#6: Line 75 – after point 4 it might also be worth mentioning that 12 month follow up is also reported.

Response:

As mentioned in earlier comments (#2 and #3.2) the main goal of this paper was to describe the co-design process for CSP and related community preferences, and hence we decided to not include the follow-up results. Only two examples (level of participation and SMS sent as feedback) have been previously included, but now have been deleted so as to improve clarity.

#7: Conceptual background: The process of co-design is well described and Figure 1 presents a clear framework.

Response:

Thank you for appreciation.

#8: Line 85 – should the references be 15-18? Otherwise there is no mention of references 16 and 17 in the conceptual background.

Response:

Reference numbers were corrected and updated throughout the manuscript.

#9: Line 105 – the should be their.

Response:

This has been corrected (Lines 111).

#10.1: Methods: The study area, study design, population and sampling is clearly described. A little further detail of the pilot workshop could be provided. How was the site for the pilot selected? Where all activities presented in subsequent workshops, undertaken in the pilot, and if so what refinements were made (if any)? Did the pilot workshop actually inform the development of the content in the subsequent workshops? Lines 213-214 indicates that the trap used was recommended from the pilot workshop. It would be good to describe this and any other outcomes/changes from the pilot.

Response:

More details explaining the methodology of the pilot study have been added to the manuscript (Lines 187-192; 212; 233-248). The pilot study was conducted in one of 10 villages where a baseline study was conducted (Asingizwe, et al. 2019), and was randomly selected. The main purpose of conducting the pilot workshop was to inform the process of the main Participatory Design Workshops (PDWs) through testing the content and steps of the PDW. Not all activities presented in PDWs were undertaken in the pilot, as some modifications were made following the pilot workshop. These included: the guiding questions, the formation of group discussion, and some steps to follow. Changes have been made to the methodology section (lines 232-296).

#10.2: Who facilitated the follow up workshops? It is stated that community members were invited to participate (line 161-162) but on line 166 it states that community members were selected by the village leaders. Did the village leaders chose three community members out of a larger number who responded to the invitation and if so what criteria was used to select the three? What was the response to the invitation like? Did each workshops have exactly the same breakdown of participants (lines 166-170)?

Response:

The first two authors facilitated all follow-up workshops, and the 6th co-author also joined the facilitation team in one of the workshops. The reporting of the process of selecting community members is revised in the manuscript. The researchers informed the village leaders about the number and the criteria of selection. At the beginning of each workshop, the researchers had to verify whether the criteria have been fulfilled. The details about selection criteria have been provided in the manuscript under the recruitment process (lines 203-226). In addition, each PDW had exactly the same breakdown of participants and followed the same steps (see recruitment process, lines 203-226).

#10.3: Line 174 – this might just be me but I have not heard the term “calling” questions. Can the authors briefly clarify what a calling question is. It seems strange to start with a question; “as a community member, how are you going to be engaged in malaria control”, before they are asked whether they are willing to participate. Was there a reason for starting with this question?

Response:

A calling question is the same as the guiding question. In the manuscript “calling” was changed to “guiding” throughout the methodology section.

This question “as a community member, how are you going to be engaged in malaria control”, was introduced right after the presentation of the results from the baseline survey (Asingizwe et al. 2019). The main reason for this question was that “citizen engagement in malaria prevention and control activities” was one of the three strategies to improve consistent use and acceptance of malaria control measures mentioned by the participants from the baseline survey in the study area. This clearly indicated that the community members were willing to participate. Consequently, we wanted the community members to think further about the process of this engagement. This reason is included in the manuscript (Lines 233-248).

#10.4: Line 183-185 – this sentence could be reworded to make it clearer.

Response:

The sentence has been revised (lines 256-258).

#10.5: I am a little confused about the questions asked in the group discussion. Line 193 states “following the calling questions” and line 197 states the groups were requested to answer “all questions”. Were the questions asked to the groups the same as the calling questions described in lines 187-191 or were they different questions? Initially I thought they were different however on line 199 the authors again refer to the calling questions so maybe they were the same. It would be good to clarify that please.

Response:

The calling questions (now revised as guiding questions) were the ones used in the group discussion. Clarification has been made in the manuscript (Lines 269-276)

#10.6: Line 200 and line 201 use the term “results” – it might be better to say they shared their “discussion” rather than shared their “results”.

Response:

This has been revised. “Results” has now been changed to “discussion” (Line 282)

#10.7: In the analysis section there is mention of flip charts being used – was this in the group discussion and if so how were they used/for what reason?

Response:

During the group discussion, participants had to write the responses to the questions on the flip charts because: (1) they had to present them and (2) the researchers wanted to keep them for further transcription of the results and analysis (lines 276-277; 314-316).

#10.8: Line 211 – how can participants give feedback on the efficacy of carbon dioxide-baited traps in collecting mosquitoes if they had not yet started the program? It sounded like these were proposed tools (line 213) states this.

Response:

We thank the reviewer for the remark. This has been changed to “perceived efficacy”(292). It is perceived efficacy based on the observed trap design and the lab results the researchers presented to the participants. Additionally, the trap was proposed for collecting mosquitoes because the majority of the participants who attended the pilot workshop reported that they could capture mosquitoes only if a tool was provided to them. Eventually, researchers anticipated the design of an efficient, low-cost trap, tested it in the laboratory and in the field and presented the results during the participatory workshops before its use (Lines 292-305)

#10.9: The paragraph lines 211-224 could be clearer.

Response:

The paragraph has been revised (Lines 292-305).

#10.10: Line 226 – the Figure should be labelled Figure 3 not Figure 2.

Response:

The figure is already labeled “Figure 3” which has parts a, b & c.

#10.11: Data analysis – how the flip charts were used needs explanation. At what point did transcription occur? It sounds like they were done during the workshop and then shared with the whole group – is that correct? How was thematic analysis actually conducted? More description is required.

Response:

The flip charts were used during group discussions (see also a response to comment #10.7). The transcription was conducted after the workshops. During the workshops, researchers had only to write a summary from the groups’ presentations and had to share it with the plenary. The main reasons of doing this were to (1) facilitate the integration session, (2) validate the responses and have a consensus as a group, (3) facilitate and inform the transcription process in case some handwritings are not readable, and (4) compliment the transcripts in case not detailed because some questions for clarifications were asked during the presentations. Clarifications have been added to the data analysis section (Lines 325-341).

The main categories of analysis were based on the guiding questions. After the completion of the analysis, the categories were divided into two main themes (technical and social) according to the co design framework (Figure 1) and the presentation of the results followed this framework. 

#10.12: In the methods section I would expect some mention of ethics – was there ethical approval for this project? Or if it was not required this should be clearly stated.

Response:

As mentioned in earlier comments, this has been added to the manuscript (lines 343-345).

#11.1: Results: Overall the results presented are interesting but as mentioned it would be good to have three clear sections for the results: The design and process of the workshop and overall project (including the pilot and the five workshops, recruiting volunteers, deciding on tools etc). The rollout (implementation) of the project. Follow up results after 12 months.

Response:

Thank you for the suggestion. However, as mentioned in earlier comments, the main objective was to present the design process and other parts have not been really included in this manuscript. Therefore, we preferred to keep the current presentation of the results. To make the results’ section more readable, an introduction indicating the flow of the results’ presentation has been added to the manuscript (Lines 348-354).  

#11.2: Line 257-260 (4.1.1): I was interested to see what participants listed as their expectations as these seemed to reflect they did not understand the purpose of the workshop (i.e. to get citizen science participation). I am not sure if this is that relevant to results and is probably not required. There is no further feedback as to whether their expectations were met.

Response:

The participants listed their expectations at the beginning of the pilot workshop and evaluated the workshop in line with their expectations at the end of the workshop. As the reviewer said we do not think this will be relevant for the current manuscript because it only presents the results related to the CSP design process. Although the expectations and evaluation steps were part of the workshops but they are not really part of the CSP. Therefore, this has been deleted (lines 360-365).

#11.3: 4.1.2: the themes do not seem to match the heading – “How participants will be engaged in malaria control”. For example how is sleeping under bed nets and early diagnosis and treatment relevant to their engagement?

Response:

This has been deleted from the manuscript (lines 366-371) and the outstanding theme “control of mosquito breeding sites” has been added under the design process of the pilot study (line 242)

#11.4: 4.1.3: Why was this question relevant? Was it to determine where interventions were required?

Response:

As the intention was to engage citizens in collecting mosquitoes, it was good to know whether participants have ever experienced mosquito nuisance. This even has been revised in the subsequent workshops where a question about the interpretation of the term mosquito nuisance was added (see the guiding question, lines 261)

#11.5: 4.1.4 is more relevant and has good content.

Response

Thank you.

#11.6: 4.2 (line 276). I would suggest starting with the characteristics of the participants and then lead into what came out of the workshop.

Response:

This has been revised and an introduction for the results section was added which implied that the characteristics of the participants are presented right after the related heading (lines 382-391).     

#11.7: Can the authors clarify what is meant when they say 185 out of 225 attended and participated in the workshop. They either attend or don’t attend. Were there RSVPs perhaps from 225 which meant they were expected to attend and only 185 turned up to the workshops. This is not clear.

Response:

225 were expected to attend because in each village we expected 45 participants. We have explained this in the manuscript (line 388).

#11.8: Line 282 – what is PDWs – I can’t see it written out in full in the manuscript but I may have missed it.

Response:

PDWs stands for Participatory Design Workshops, and as mentioned in earlier comment (#6.b) the definition has been added in the manuscript (lines 182-186).

#11.9: Line 311 – what is PW – I can’t see it written out in full in the manuscript but I may have missed it. Further clarity is required in the sentences on lines 311-317. The PW trial needs explanation. I am unclear if the suggestion to use the carbon dioxide baited traps came out of the pilot workshop or the PW trial.

Response:

The reviewer is correct, the suggestion to use the carbon dioxide baited traps came out of the pilot workshop, and to remove confusion PW (participatory workshop) has been deleted from the manuscript and replaced by the pilot workshop (line 419)

#11.10: Line 336 – no need to write community health workers out in full as this has been abbreviated previously.

Response:

This has been corrected in the revised manuscript and only abbreviation has been maintained (line 444).

#11.11: From Section 4.2.3 onwards is where I think clearer delineation between the design and the roll out (implementation) of the citizen science program can be made.

Response:

As mentioned in the previous comments (#2 and #3.2) the manuscript presented the design process of the citizen science program. Additionally, other results (level of participation, and examples of SMS) has been deleted to prevent further confusion (lines 462-463; 562).

#11.12: On line 354 it is stated that there was 93% participation after nine months. This could come after the implementation is presented and detail about the methods to assess this are required. How was participation measured.

Response:

This level of participation has been deleted from the manuscript as it is not part of the design process. Please also see our response to earlier comments (#2 and #3.2).  

#11.13: Line 366-367: I am surprised by the females concerns – would they not be trained and given materials as part of being involved in the citizen science program?

Response:

The groups did not know whether materials would be offered or not because we wanted to use locally available materials so that the participants may even look for those materials on their own. The main reason for this was the sustainability part of the program, because initially, we wanted the participants to find a method to collect mosquitoes by themselves that fit their context and needs, so that even at the end of the research and funding, the program can continue running.

#11.14: Line 376 – this should be Figure 5 not Figure 4. I am not sure what Figure 4 on lines 378-394 is as it is not referred to in the text. It is also incorrectly numbered.

Response:

This is Figure 4 and not figure 5, and that is the one referred in the text (line 484). We do not think there was inconsistency in numbering figures unless something went wrong through downloading the documents.

#11.14: Lines 414-425: It is stated that reporting happened once a month but in Table 2 some report fortnightly (women in one group and youth in one group) or weekly (youth in one group).

Response:

It is true that there was some variation in the frequency of reporting the observations; some indicated once a month, others once in two weeks, or once a week (line 544). However, we had to agree on a final decision (choice) because it is better to have the same reporting time for the researchers to be able to analyze, compare, and provide feedback.

#11.15: Lines 440-441 and Table 4: I am confused where these preferences /choices came from – was it after the workshop or after the roll out of the program?

Response:

Table 4 (which changed to table 3 after table 3 has been deleted) summarizes the results (design choices) from the PDWs (line 566). In addition, as mentioned in earlier comments (#2 and #3.2), the results of the roll out are not presented.

# 12.1: Discussion: The discussion could be clearer and may well be following some modifications to how results are presented. Overall it would be good to see the main focus of the discussion on the co-design process and what can be learnt from that. Learnings from the implementation will also be of interest and this seems to covered in lines 484-497.

Response:

As the results were not changed and they are presented following the design process presented in Figure 1; therefore, the structure of the discussion also was not changed as it also follows the same flow under two main headings: (1) involving citizen in the design process and (2) ways of feedback provision. The information presented in the latter does not belong to implementation because it showed how the feedback was considered in the design process and the importance of this feedback. The discussion section about the co design has been extended and the level of participation was deleted (lines 574-597; 604-606; 609-611).

#12.2: There is content in the discussion on line 452 and 453 that contains information that is not clearly presented in the results – i.e. hardly any variation in participation throughout the first year of the program.

Response:

This discussion was based on the level of participation presented in the results (around 93% throughout the year). However, as we deleted this information from the results, consequently, this part of the discussion has been deleted as well (580-590).

#12.3: I am not sure how relevant the content is that is described in lines 453-461 but rewriting may improve this clarity and relevance.

Response:

This was indeed related to the level of participation presented in other citizen science studies, however, this was deleted as the corresponding results were deleted as well (580-590).

#13.1: Study limitations and future research: This section could be refined.

Response:

This section has been revised and some information (for example information related to why paper based form has been preferred over the mobile phone) has been added (Lines 635; 651-654; 668). 

#13.2: Line 500-501: When the authors refer to well-documented rules and regulations – can these be explained. Do you mean clear guidelines/instructions?

Response:

By well-documented rules and regulations, we partially referred to clear guidelines, but we also referred to the clear actions which may prevent people to deviate from normal protocols and procedures. This has been added to the manuscript (line 635).

#13.3: Line 509: I am not sure if you can call motivation a key scientific reason. Just needs rewording.

Response:

This has been revised to “an important reason” (line 644).

#13.3: I think an important recommendation should be about the need for rigorous evaluations of citizen science programs.

Response:

Thank you for the suggestion. This was already included in the manuscript (lines 662-668), and it has been also revised and included the evaluation term.

#14: Conclusion: This may need to be refined following a rewrite of the results and discussion.

Response:

This has been slightly revised and the results which were not presented have been deleted.

#15: References: Up-to-date references used. A few minor presentation corrections required.

Response:

References has been revised accordingly.

Reviewer 2 Report

Due October 31, 2019

Review request: Sustainability

Manuscript ID: Sustainability-631029

Title:    Co-designing a citizen science program for malaria control in Rwanda

Synopsis

Rwanda has struggled to maintain funding to support consistent and sustainable mosquito surveillance programs. The involvement of citizens in a scientific collaboration program to meet community needs (malaria threats) and to collect local data may help in sustaining a consistent surveillance system. Line 70-72 state, “This paper outlines how such a surveillance program could be designed, put in place and what preferences exist in local communities with regard to the technical and social components of such an approach.” This citizen science program incorporated participatory action research processes successfully and built context-specific knowledge and skills for participants in rural African communities. This is the first-year report of the program.

Reviewer's conflict of interest: None

Comment to authors

A well-written manuscript. My comments are as follows:

Title. Minor suggestions: 1) as more publications are expected to follow, you can supplement information about the progression of a program by adding the dimension of time, such as “Co-designing……. In Rwanda: A year one report” or “…: 2017-2019”; and 2) “citizen science program” seems to be a key name and appears 23 times across the manuscript. Can it be capitalized as “Citizen Science Program” (CSP)?

Abstract. This CSP is based on a citizen science approach (CSA) (line 22). Participatory action research (PAR) is a type of research methodology used since the 1940s. CSA is a separate variation to promote scientific knowledge and skills among community people. An important distinction can be seen in Africa arising from the fact that many African villages may still be dominated by animistic beliefs which may end up being augmented by a simple application of PAR. CSA has the special purpose of teaching science rather than directly applying PAR. Is it possible to add the objective of CSA and to explain how a simple application of PAR may be problematic by reinforcing unscientific beliefs?

Introduction. Well-written paragraphs explaining the existing system and the citations are coherent; however, the citation number should appear at the end of a sentence as a reference to support the sentence. For example [5] in line 47 should appear at the end of the sentence on line 48. An apostrophe is missing in line 46 “United Nations’ sustainable development goals”. The gap in knowledge and the purpose statements are well presented.

Conceptual Background. This is also a part of the introduction and it is good to have this subheading.

Methods. Recruitment method was not clear. In participatory action projects, the recruitment and/or outreach is one of the key components of success, because it can draw sufficient attention to the community and recruit key individuals from different stakeholder groups. Although this was not a simple PAR, please describe a proposed method, as well as the results of the recruitment and outreach methods.

As stated above, please include a paragraph about recruitment resulting in the range of 38 – 100% attendance at six workshop locations (Table 1). Line 282, “PDW” (participatory or pilot design workshops?) has not been defined. Table 1: Why did Busasamana have low attendance? Is the “45” target attendance based on the population of each village? If possible, I would like to know about the ratio of reporting by paper versus reporting by SMS, and the SMS capability at each village. This ratio may indicate the technological advancement (accessibility of internet and cell phones) for later report comparisons. However, in the limitations section, people did not prefer using SMS due to the cost (line 473). Will subsidizing the cost allow them to use the SMS technology? If this issue is too complex, a current summary report will be fine. Is “Isibo” municipally defined or a naturally forming small group of households and how is a leader selected in each Isibo? Please add an explanation on line 163 because this particular information is not available online and further confused by the description of Isibo leaders and representatives (Line 371 – 376). The right side parenthesis is missing for “(“ in line 372. Figure 3 is confusing because each village has one Isibo leader. My understanding was a village is composed of multiple Isibos. Is Isibo synonymous with a village? Table 3 and 4 are well presented.

Discussion. All paragraphs are well-written. With regard to the “well-documented rules and regulations” (line 500), ethics in science and technology is particularly important in sustainability of any well-intended programs. False reports and inability to provide accurate data will lead to misallocation of resources which may lead to inequality and systematic bias. Such misallocation of resources can happen with or without clear intention of participants. It will be beneficial to include paragraphs that researchers are aware of ethical issues.  

Conclusion. Good.

Additional information. Please include the disclaimer of exempt from human subject research regulations due to the public health nature of the project.

The format of few references—bold letter for published year—is inconsistent. Please check the format of each reference.

Overall, I believe this paper presents an important progression of science-based participatory action research. The exemplar is practical and representative in order to allow a community project to become sustainable.

End of review.

Author Response

Reviewer 1

A well-written manuscript. My comments are as follows:

#1 a: Title. Minor suggestions: as more publications are expected to follow, you can supplement information about the progression of a program by adding the dimension of time, such as “Co-designing……. In Rwanda: A year one report” or “…: 2017-2019”;

Response:

As the manuscript is dedicated to the process to undergo while co-designing and implementing the citizen science program, and not reporting on the results of the first year of the implementation, we would prefer to leave out the dimension of time.

#1 b: “citizen science program” seems to be a key name and appears 23 times across the manuscript. Can it be capitalized as “Citizen Science Program” (CSP)?

Response: 

We agree. The changes were made for the “Citizen Science Program (CSP)” throughout the manuscript, and the title was capitalized as “Co-designing a Citizen Science Program for Malaria Control in Rwanda” (Lines 2-3).

#2: Abstract. This CSP is based on a citizen science approach (CSA) (line 22). Participatory action research (PAR) is a type of research methodology used since the 1940s. CSA is a separate variation to promote scientific knowledge and skills among community people. An important distinction can be seen in Africa arising from the fact that many African villages may still be dominated by animistic beliefs which that may end up being augmented by a simple application of PAR. CSA has the special purpose of teaching science rather than directly applying PAR. Is it possible to add the objective of CSA and to explain how a simple application of PAR may be problematic by reinforcing unscientific beliefs?

Response:

Indeed, PAR is a research methodology that has previously been described in the literature. However, the difference between CSP and PAR is that in CSP, citizens are actively engaged in the collection, and or analysis and interpretation of the data, while PAR or other bottom-up/community-based interventions involve people who are affected by a particular related event and include them as key partners in the discussion and problem-solving cycle. In other words, for PAR, there is a lack of active engagement, in sharp contrast to co-designed CSP. This co-creation in CSP also enhances ownership which may stimulate sustainability in the long run. However, with PAR, when the researchers step out and external resources are over, then the project, most of the time, would also end. We added a distinction between CSA and PAR in the conceptual background (Lines 86-90). However, as this paper is about CSP we do not think it is necessary to explain how a simple application of PAR may be problematic by reinforcing unscientific beliefs mainly because we are not comparing the two approaches (methodologies).

#3: Introduction. Well-written paragraphs explaining the existing system and the citations are coherent; however, the citation number should appear at the end of a sentence as a reference to support the sentence. For example [5] in line 47 should appear at the end of the sentence on line 48. An apostrophe is missing in line 46 “United Nations’ sustainable development goals”. The gap in knowledge and the purpose statements are well presented.

Response:

The citation [5] is moved to the end of the sentence. The apostrophe is also added (Lines 47-48).

#4: Conceptual Background. This is also a part of the introduction and it is good to have this subheading.

Response:

Thank you.

#5: Methods. Recruitment method was not clear. In participatory action projects, the recruitment and/or outreach is one of the key components of success, because it can draw sufficient attention to the community and recruit key individuals from different stakeholder groups. Although this was not a simple PAR, please describe a proposed method, as well as the results of the recruitment and outreach methods.

Response:

We have added a section providing the details about the recruitment process (Lines 201-226). Based on ten villages selected in the baseline survey that was exploring the baseline information (Asingizwe et al., 2019), five of these (one village per cell) were selected for the implementation of the CSP. On average each village has 150 households and we targeted a third (45 community members). The households are grouped in “isibo” (a cluster of a maximum of 15 neighboring households), thus each village has approximately 10 isibo. Therefore, three community members in each isibo and the isibo leader were recruited to participate which gives a total of 40 participants per village. In addition, each village has three Community Health Workers and one village leader. Consequently, these were also added to 40 community members selected. Lastly, an executive of the cell was also invited to attend the workshop. Hence, in total 45 people were recruited in each of the five villages selected. The village leaders selected the community members and we had to make sure that these community members are not from the same household, or relatives. To ensure this, the village leader had to announce this workshop during a village meeting and those who showed interest were invited to participate. As shown in table 1, in some villages community members were not motivated to attend in a good number. The main reason for this low turn up in these villages is that these villages (Busasama and Kibaza) are located far from the health center where the workshops were conducted. In addition, the day we conducted a workshop for Busasamana it was raining, and some community members decided to go into their farms for fieldwork instead of attending the workshops.

#6: Results. As stated above, please include a paragraph about recruitment resulting in the range of 38 – 100% attendance at six workshop locations (Table 1). Line 282, “PDW” (participatory or pilot design workshops?) has not been defined. Table 1: Why did Busasamana have low attendance? Is the “45” target attendance based on the population of each village? If possible, I would like to know about the ratio of reporting by paper versus reporting by SMS, and the SMS capability at each village. This ratio may indicate the technological advancement (accessibility of internet and cell phones) for later report comparisons. However, in the limitations section, people did not prefer using SMS due to the cost (line 473). Will subsidizing the cost allow them to use the SMS technology? If this issue is too complex, a current summary report will be fine. Is “Isibo” municipally defined or a naturally forming small group of households and how is a leader selected in each Isibo? Please add an explanation on line 163 because this particular information is not available online and further confused by the description of Isibo leaders and representatives (Line 371 – 376). The right side parenthesis is missing for “(“ in line 372. Figure 3 is confusing because each village has one Isibo leader. My understanding was a village is composed of multiple Isibos. Is Isibo synonymous with a village? Table 3 and 4 are well presented.

Response:

This comment is subdivided into small section below in order to address it clearly. 

#6 a: Results. As stated above, please include a paragraph about recruitment resulting in the range of 38 – 100% attendance at six workshop locations (Table 1).

Response:

A recruitment process was added in the methodology to explain the differences observed in the attendance (Lines 201-226).

#6 b: Results. Line 282, “PDW” (participatory or pilot design workshops?) has not been defined.

Response:

A definition of Participatory Design Workshop (PDW) is added (Lines 182-186). PDW in this context is a workshop through which all stakeholders including users in this case citizens affected by the upsurge of malaria in their environment are invited to define together the problem that affects them and to set up mechanisms to solve the problem while anticipating their needs. It is therefore a user-centered design method where the focus is on the active role of users.

#6 c: Results. Table 1: Why did Busasamana have low attendance?

Response:

The reason why Busasamana had low attendance is explained in the recruitment process section (Lines 221-226). The main reasons are that Busasamana is located far from the health center where the workshops were conducted. In addition, the day we conducted a workshop it was raining, and some community members decided to go into their farms for field work than attending the workshop (see also a reply for comment #5).

#6 d: Results. Is the “45” target attendance based on the population of each village?

Response:

As mentioned also in the reply of comment #5, 45 targeted participation was in proportion of the size of the villages (number of households). Approximately each village has 150 households.

#6 e: Results. If possible, I would like to know about the ratio of reporting by paper versus reporting by SMS, and the SMS capability at each village. This ratio may indicate the technological advancement (accessibility of internet and cell phones) for later report comparisons.

Response:

The paper based form was designed and proposed after the results of the baseline collected in 2017, where only 45% in the study area owned the mobile phone and among these only a small proportion (18%) mentioned that they also use their mobile phone for SMS activities (sending and or receiving any message) (Unpublished data). To this end, proposing a paper based form was a way to limit the technology related barrier. As suggested this is included in the limitation section (lines 651-654).

As reported in the results section (Lines 399-416), the participants opted to use paper-based forms over SMS due to cost involved in sending SMS. Therefore, there is no participant who reported the observations using SMS, but instead each participant filled out the paper form, and the paper forms from each participant were submitted to respective isibo leader in turn who submitted the forms to the researchers. After the entry of the data into an Excel spreadsheet, the data were analysed. Researchers shared the results via SMS with the isibo representatives on a monthly basis and this has no cost implications for the receiver of this SMS.

#6 f: Results.  However, in the limitations section, people did not prefer using SMS due to the cost (line 473). Will subsidizing the cost allow them to use the SMS technology? If this issue is too complex, a current summary report will be fine.

Response:

As mentioned in the previous response, seeing the limitations that SMS imposed, paper based form was used to submit the report, and Researchers sent an SMS to the isibo representative on a monthly basis as feedback and this does not have any cost to the receiver of the SMS (Lines 546-557; 604-606).

#6 g: Results.  Is “Isibo” municipally defined or a naturally forming small group of households and how is a leader selected in each Isibo?

Response:

Isibo is defined as a small group of households. The isibo representative was selected by the volunteers for the purpose of this program (Lines 206)

#6 h: Results. Please add an explanation on line 163 because this particular information is not available online and further confused by the description of Isibo leaders and representatives (Line 371 – 376).

Response:

Explanation of isibo is added (Line 206) and the distinction between isibo leader and isibo representative is given (206, 482-528). Leaders are those selected by community members and in charge of 15 households neighboring each other, while representatives are those who were selected by the volunteers for the purpose of the Citizen Science Program (one representative per village).

#6 i: Results. The right side parenthesis is missing for “(“ in line 372.

Response:

The right side parenthesis is added (Line 481).

#6 j: Results. Results. ….. Figure 3 is confusing because each village has one Isibo leader. My understanding was a village is composed of multiple Isibo. Is Isibo synonymous with a village? Table 3 and 4 are well presented.

Response:

For Figure 3, (I guess here is figure 4 because it is the one presenting the information about isibo and village). The reviewer is right, normally, a village is composed of multiple isibos (approximately 10 on average). This confirms that isibo is not synonymous to a village. The caption of figure 4 was corrected and it indicates the isibo representatives not isibo leaders (Line 515). However, as we have a certain number of volunteers in each village, consequently these form one group (a group of volunteers) in each village and have one representative respectively. The representatives of the volunteers for the citizen science program are called “isibo representatives” because were selected among other isibo leaders. The black arrow (figure 4) shows that these isibo representatives are directly connected to researchers because they are the ones submitting the observations every last Friday of the months  (Lines 518-520; 521-529).

#7. Discussion. All paragraphs are well-written. With regard to the “well-documented rules and regulations” (line 500), ethics in science and technology is particularly important in the sustainability of any well-intended programs. False reports and inability to provide accurate data will lead to misallocation of resources which may lead to inequality and systematic bias. Such misallocation of resources can happen with or without clear intention of participants. It will be beneficial to include paragraphs that researchers are aware of ethical issues.  

Response:

The reviewer is right by pointing out that an ethical statement should be added. In fact, this research was approved by the Institutional Review Board of the College of Medicine and Health Sciences, University of Rwanda. Consequently, a section about ethical approval was added in the manuscript (Lines 343-345).

8. Good.

Response:

Thank you.

#9. Additional information. Please include the disclaimer of exempt from human subject research regulations due to the public health nature of the project.

Response:

A section about ethical approval from the Institutional Review Board of the College of Medicine and Health Sciences, University of Rwanda has been added (Lines 343-345). Also see our response to comment #7.

#10. The format of few references—bold letter for published year—is inconsistent. Please check the format of each reference.

Response:

The references are now revised in line with the journal’s format.

#11. Overall, I believe this paper presents an important progression of science-based participatory action research. The exemplar is practical and representative in order to allow a community project to become sustainable.

Response:

Thank you for your appreciation.

Reviewer 3 Report

This study used co-design process to develop a citizen science program for malaria control in Rwanda and is written for a special issue on 

"Citizen Science and the Role in Sustainable Development".

I think the manuscript is very suitable for the issue since the research design allows the researchers and participants to mutually work on the project of malaria control. Although the numbers of Figures and Tables in this manuscript are many, Figure 1 is very helpful for readers to understand what is the co-designing process of the citizen science program. On the other hand, I am puzzled if we really need Figure 5. Figure 5 is the statement of part of the results. It says the "process" of reporting the observation on a monthly basis. Is this statement placed to be under Method? Similarly, Table 3 may not necessarily be Table. The authors may write down text in Result section. 

This study is based on the 6 times of workshop where the participants vary in each time so the results may be biased. In addition, given that Kagasera village had small percentages of men, I am wondering it might influence the findings. 

Author Response

Reviewer 2

Comments and Suggestions for Authors

This study used co-design process to develop a citizen science program for malaria control in Rwanda and is written for a special issue on "Citizen Science and the Role in Sustainable Development".

#1: I think the manuscript is very suitable for the issue since the research design allows the researchers and participants to mutually work on the project of malaria control. Although the numbers of Figures and Tables in this manuscript are many, Figure 1 is very helpful for readers to understand what is the co-designing process of the citizen science program. On the other hand, I am puzzled if we really need Figure 5. Figure 5 is the statement of part of the results. It says the "process" of reporting the observation on a monthly basis. Is this statement placed to be under Method? Similarly, Table 3 may not necessarily be Table. The authors may write down text in Result section. 

Response: 

As suggested by the reviewer, Figure 5 was removed from the manuscript and we maintained the text explaining the process of reporting (521-533). We thank the Reviewer for the suggestion to place this statement about the process of reporting the information under the method section, however, this is a methodological paper in which this process was discussed in the workshops and it is the results of the discussion. Therefore, the process indicates the steps to follow throughout the reporting circle. For this reason, we prefer to maintain this information in the results section.

For Table 3, we thought that the information presented would be clearer in a table than in the text alone, because the messages were sent in the local language “Kinyarwanda”, and translated versions are provided. However, as pointed out by reviewer #3, this table was removed because it presented some of the information related to the implementation phase.  

#2: This study is based on the 6 times of workshop where the participants vary in each time so the results may be biased. In addition, given that Kagasera village had small percentages of men, I am wondering it might influence the findings.

Response:

The five participatory design workshops were conducted to agree on the tools to use and the process to follow, and another participatory design workshop was conducted as a pilot. The biases were minimized, because although the participants are from different villages, they have common characteristics (they share the same geographical area), and the steps for each workshop (Figure 2) were the same.

2. We agree that in Kagasera, and even in general, more women attended the workshops. However, during the discussion, we created homogenous groups (consisting of women, men, and youth) to ensure that all participants have equal opportunities to share their views (Lines 270-274). Consequently, ideas from each group were presented and discussed for approval.

Round 2

Reviewer 1 Report

Thank you for the opportunity to re-look at your paper following your responses to the reviewers' feedback. You have provided a detailed response to each of the reviewers' comments and as a result the paper is now much stronger. A few minor editing corrections are required - for example:

Make sure you are using the CSP abbreviation in all parts of the document - see lines 572, 578, 597 and 618.

Line 577 - "making" decisions rather than "taking".

Line 579 - "these" incentives rather than "this".

Author Response

Reviewer one

Thank you for the opportunity to re-look at your paper following your responses to the reviewers' feedback. You have provided a detailed response to each of the reviewers' comments and as a result the paper is now much stronger. A few minor editing corrections are required - for example:

#1: Make sure you are using the CSP abbreviation in all parts of the document - see lines 572, 578, 597 and 618.

Response:

The CSP has been now used throughout the manuscript.

#2: Line 577 - "making" decisions rather than "taking".

Response

Now "taking" has been changed into "making" (line 577)

#3: Line 579 - "these" incentives rather than "this".

 Response:

Now "this" has been changed into "these" (line 579)

Reviewer 3 Report

The authors sufficiently responded to my previous comments.

Author Response

Reviewer three

The authors sufficiently responded to my previous comments.

Response:

Thank you